# Resistance Training Improves Muscle Strength and Function, Regardless of Protein Supplementation, in the Mid- to Long-Term Period after Gastric Bypass

**DOI:** 10.3390/nu14010014

**Published:** 2021-12-21

**Authors:** Gabriela S. Oliveira, Flávio T. Vieira, Fernando Lamarca, Ricardo M. Lima, Kênia M. B. Carvalho, Eliane S. Dutra

**Affiliations:** 1Graduate Program in Human Nutrition, University of Brasília, Brasília 70910-900, Brazil; gabriela.17oliveira@gmail.com (G.S.O.); flavio.nut@hotmail.com (F.T.V.); fernando.lamarca@unirio.br (F.L.); professorricardomoreno@gmail.com (R.M.L.); kenia@unb.br (K.M.B.C.); 2Department of Applied Nutrition, Federal University of the State of Rio de Janeiro, Rio de Janeiro 22290-240, Brazil; 3Graduate Program in Physical Education, University of Brasília, Brasília 70910-900, Brazil

**Keywords:** bariatric surgery, muscle strength, physical function, resistance training, Roux-en-Y gastric bypass, whey protein

## Abstract

Inadequate protein intake and low levels of physical activity are common long-term sequelae after bariatric surgery and can negatively affect muscle strength (MS) and physical function (PF). The study investigated the effects of resistance training with or without protein supplementation on MS and PF. The study, which involved a 12-week controlled trial (*n* = 61) of individuals 2–7 years post-Roux-en-Y gastric bypass (RYGB), comprised four groups: whey protein supplementation (PRO; *n* = 18), maltodextrin placebo (control [CON]; *n* = 17), resistance training combined with placebo (RTP; *n* = 11), and resistance training combined with whey protein supplementation (RTP+PRO; *n* = 15). An isokinetic dynamometer was used to measure MS (peak torque at 60°/s and 180°/s). PF was measured with the 30-s sit-to-stand (30-STS), 6-min walk (6-MWT), and timed up-and-go (TUG) tests. There were improvements in the absolute and relative-to-bodyweight peak torque at 60°/s and 180°/s, TUG, 6-MWT and 30-STS in the RTP and RTP+PRO groups, but not in the CON and PRO groups. Changes in MS were significantly correlated with changes in PF between the pre- and post-intervention periods. A supervised resistance training program, regardless of protein supplementation, improved MS and PF in the mid-to-long-term period after RYGB and can lead to clinical benefits and improved quality of life.

## 1. Introduction

The limited success of nonpharmacological and pharmacological therapies in patients with obesity has led to a rapid increase in the number of bariatric surgeries (BSs) over the last two decades [1]. Some positive aspects related to BS include efficient long-term weight loss for severe obesity, remission of obesity-related comorbidities [2,3,4], and a decrease in the risk of mortality [2,5]. Worldwide, one of the most frequently performed procedures is the Roux-en-Y gastric bypass (RYGB) [6], which is a restrictive and malabsorptive procedure [7].

The profound effect of RYGB in decreasing fat mass is well documented; however, people who have undergone BS, such as RYGB, also present with a significant decrease in fat-free mass (FFM), particularly the skeletal muscle [8,9], which can negatively affect muscle strength (MS) and the ability to perform activities of daily living [2,10,11]. People with greater MS who underwent RYGB had faster gait speed and chair stand time, while MS significantly decreased from preoperative values [12]. Thus, strategies to improve MS and function after RYGB are required.

Exercise training following RYGB improves MS and physical fitness after BS-induced weight loss [13]. However, information about these effects in the long term after BS is scarce, since most of the randomized controlled trials in this topic focus on the early postoperative period [13]. It has been postulated that protein supplementation enhances resistance training effects [14,15], but data on protein supplementation and RYGB are limited. Oppert et al. demonstrated that resistance training combined with protein supplementation prevented the loss of MS after BS, but the study lacked a resistance training group without supplementation [11]. Moreover, to our knowledge, no previous studies have examined the effects of resistance training, with or without protein supplementation, in the mid- to long-term (>2 years) period after RYGB, for which more data are required, since weight is typically regained and physical function (PF) may consequently be compromised [16].

Therefore, this study aimed to investigate the effects of resistance training, with or without whey protein supplementation, on MS and PF in people 2–7 years after RYGB.

## 2. Materials and Methods

### 2.1. Study Design and Ethical Approval

This study was part of the Nutrition and Resistance Exercise in Obesity (NERO) project, which was a controlled clinical trial with parallel groups (RBR-9k2s42/). The study protocol was approved by the Research Ethics Committee of the Faculty of Health Sciences, University of Brasilia (nº 2.052.734, 9 May 2017).

### 2.2. Participants

Participants of both sexes were considered eligible if they were 18–60 years at 2–7 years after RYGB. We excluded individuals who regularly used protein supplements; were engaged in physical exercise for at least 2 months before the study; had diabetes mellitus, heart disease, severe psychiatric disorders, or any decompensated chronic disease, as well as malignant or wasting disease; were using hormones or appetite regulators; or who had an amputation, were pregnant, or breastfeeding.

### 2.3. Allocation

The study protocol has been previously described [17]. Briefly, participants (*n* = 119) were partially matched for body mass index (BMI), age, sex, and postoperative time. The participants were allocated to the resistance training program considering their time, transportation availability, and ability to reach the place where it was being conducted, which was located far from the city center and was difficult to access. For supplementation, the allocation was performed using random and double-blind procedures (Research Randomizer^®^ online software, version 4.0 http://www.randomizer.org/) accessed on 8 September 2017. Volunteers were allocated to one of four possible study groups as follows: whey protein supplementation (PRO), maltodextrin placebo (control [CON]), resistance training combined with placebo (RTP), and resistance training combined with whey protein supplementation (RTP+PRO).

### 2.4. Sociodemographic, Clinical, and Anthropometric Characteristics

A questionnaire regarding sex, date of birth, educational level (in years of study), and surgery date (in months and years) was administered. Anthropometric measurements were assessed using standard procedures. Weight loss was considered satisfactory when excess weight loss and total weight loss exceeded 50% and 20%, respectively. Weight regain was considered present if the current weight was greater than 10% of the lowest postoperative reported weight.

### 2.5. Resistance Training Program

Before the training intervention, volunteers underwent three familiarization sessions to ensure that they followed the proper exercise techniques. The resistance training program was performed three times per week on non-consecutive days for 12 weeks. Each training session was preceded by a 10-min warmup and lasted approximately 60 min. The protocol consisted of the following eight exercises targeting all major muscle groups: chest press, knee extension, hamstring curl, leg press, hip abduction, lat pulldown, shoulder abduction, and plantar flexion (Rotech^®^ Fitness Equipment, Goiânia, Brazil).

Training loads were monitored and adjusted using the OMNI Resistance Exercise Scale (OMNI-RES) [18]. The program followed a progressive trend, with training loads at 6 points (relatively hard) during the first 4 weeks, 7 points (between relatively hard and hard) during the following 4 weeks, and 8 points (hard) over the remaining 4 weeks, with repetitions decreasing from 12, 10, to 8 over each 4-week period. All exercises were performed in three sets with approximately 1-min rest intervals between sets and exercises. In each session, participants were individually supervised by qualified professionals to precisely adjust loads and verify proper execution. Volunteers were instructed to avoid altering their daily habitual routines throughout the entire study period. Participants who did not attend at least 70% of the training sessions were excluded from the data analysis.

### 2.6. Nutritional Intervention

The amount of concentrated whey protein or maltodextrin powder provided to the corresponding groups was 0.5 g/kg of ideal bodyweight/day. The ideal weight was calculated considering an ideal BMI of 25 kg/m^2^. This proposed protein amount was based on a supplementation protocol previously suggested by a randomized controlled trial that investigated the effect of dietary whey protein supplementation on the weight loss and body composition changes of women who regained weight 24 or more months after RYGB [19]. Furthermore, according to clinical practice guidelines for the perioperative nutrition, metabolic, and nonsurgical support of patients undergoing bariatric procedures (2020) [20], this amount provides one-third of the daily recommendation of 1.5 g protein/kg ideal weight per day. The nutritional composition of concentrated whey protein (30 g) corresponded to 120 kcal, 1.80 g of carbohydrates, 1.38 g of total fats, and 23.10 g of proteins (5.61 g of branched-chain amino acids, which included 2.70 g of leucine). The same portion of non-flavored maltodextrin provided 112 kcal and 28 g of carbohydrates.

Whey protein or placebo, prepackaged in the amount corresponding to the individually calculated daily dose, was distributed every 15 days. As the colors of the products were different, the supplements were delivered in opaque packaging. The instructions for supplement consumption, provided to all participants, consisted of diluting the daily dose, which was to be taken in a single portion together with the last meal of the day, without replacing it. Any amount not consumed was to be returned during scheduled follow-up consultations to assess supplementation adherence. Participants whose intake was less than 70% of any of the provided supplements were considered lost to follow-up and were excluded from the data analysis.

### 2.7. Isokinetic Assessment

An isokinetic dynamometer (Biodex 3, Biodex Medical, Inc., Shirley, NY, USA) was used to measure the participants’ dominant knee extensor peak torque (PT) and work capacity. After familiarization, individuals underwent a testing protocol of two sets of four repetitions at 60°/s, followed by two sets of four repetitions at 180°/s, with 60 s of rest between all sets [21]. The highest elicited PT at each angular speed, which was considered as an absolute value (Nm) and relative-to-bodyweight value (Nm/kg), was used for subsequent analyses. Work capacity was calculated as the amount of torque produced throughout the entire range of motion for all repetitions. For the procedures, volunteers were positioned on the dynamometer seat with belts fastened at the trunk, pelvis, and thigh to avoid any compensatory movements that could affect the results. The lateral epicondyle of the femur was identified to align it with the dynamometer rotation axis. Gravity correction was achieved by measuring the torque exerted by the lever arm and the subject’s leg at 30° flexion in a relaxed position. A trained evaluator provided verbal encouragement to the participants while they performed the movements with their maximal strength. Equipment was calibrated according to the manufacturer’s specifications before each evaluation session. In our laboratory, the test-retest reliability coefficient value for knee extensor PT was 0.91.

### 2.8. Physical Function

The timed up-and-go test (TUG), 6-min walk test (6-MWT), and 30-s sit-to-stand test (30-STS) were conducted as measures of PF.

The TUG procedures were fully explained before the assessment, after a familiarization trial. Briefly, volunteers were individually seated in a standard chair with a height of 46 cm, with their back against the chair, both arms resting alongside the body, and both feet completely resting on the floor at a comfortable distance. Participants were instructed to, on the word “go,” get up and walk 3 m forward, as fast as possible, turn around an obstacle, return to the chair, and sit down again. The best time was recorded after three attempts, with 60 s of rest between attempts [22].

The 6-MWT was conducted using a circuit that was 45.72 m in length, with cones placed every 4.57 m. Participants were instructed to walk at their own pace to cover as much distance as possible in 6 min, without running. The covered distance, in meters, was considered for the analyses [23].

The 30-STS was conducted with a standard (46-cm high) chair. With their arms crossed over their chest, participants were instructed to stand up fully and sit down fully as many times as possible within 30 s. Only the complete repetitions were recorded [23].

### 2.9. Dietary Intake

Two 24-h dietary recalls, one in person and one by telephone on non-consecutive days, as well at 12 weeks of intervention, were applied by qualified evaluators at baseline. The interviews were conducted following the 5-step multiple-pass method [24] to stimulate the respondent’s memory and increase the accuracy of the reported information [25]. The Nutrition Data System for Research (NDSR^®^) software, version 2018 (Nutrition Coordinating Center, University of Minnesota, Minneapolis, MN, USA) was used to analyze the collected dietary intake information. The usual intake was estimated using PC-SIDE^®^ software (version 1.0; Iowa State University, Ames, IA, USA).

### 2.10. Statistical Analysis

The Kolmogorov–Smirnov test was applied to verify the normality of data distribution. Homogeneity of variance was analyzed using Levene’s test. Between-group comparisons at baseline and food intake after 12 weeks were performed using one-way ANOVA with Tukey’s post hoc test. Kruskal-Wallis H tests were performed to evaluate the delta values. A per protocol analysis was performed using a two-way mixed ANOVA test with repeated measures for time × group interactions and Fisher´s Least Significant Difference post hoc test were performed to analyze the effects of isolated and combined study interventions on MS and PF variables. Time was considered an intra-individual factor, while group was an inter-individual factor. Correlations between MS and PF were analyzed using Spearman’s test. Statistical significance was set at *p* < 0.05. All analyses were performed using SPSS software (version 25.0; IBM Corp., Armonk, NY, USA).

## 3. Results

A total of 119 subjects were first allocated to the study groups. However, 58 participants (48.7%) did not complete the clinical trial. The loss to follow-up occurred due to a lack of adherence to placebo or whey protein supplementation (*n* = 8), dermolipectomy (*n* = 2), illnesses such as musculoskeletal problems, familial diseases, and gout, or accidents, such as car accidents (*n* = 9), nonspecific edema (*n* = 1), pregnancy (*n* = 1), involvement in an external exercise program (*n* = 1), and decline (*n* = 34). Two volunteers had missing baseline MS or PF data; therefore, they were excluded from per protocol analysis (Figure 1).

Generally, those who remained in the study were mostly women, with an approximate mean age of 40 years, BMI of 30 kg/m^2^, and post-surgery duration of 4 years. There were no significant differences in characteristics between the participants who remained and those who left the study (data not shown). Although most participants exhibited adequate weight loss, almost half of them regained weight (Table 1). The total energy and protein intake at baseline were approximately 20 kcal/kg of the current weight and 70 g/day, respectively. During the follow-up, these values were similar between the groups (*p* > 0.05), except for the PRO group, which had a higher protein intake than the CON group after 12 weeks of intervention. The mean adherence to the protein supplementation and the resistance training program was greater than 90% and 80%, respectively (Table 2).

No training-related events occurred during the testing or training sessions. Significant time × group interactions were observed for PT at 60°/s, PT at 60°/s relative to the body weight, total work at 60°/s, PT at 180°/s, PT at 180°/s relative to the body weight, total work at 180°/s, TUG, 6-MWT, and 30-STS (Table 3). Post hoc analyses revealed that these interactions were driven by significant improvements in the exercise groups (i.e., RTP and RTP+PRO), while the non-exercise groups (i.e., CON and PRO) did not show any significant alterations between the pre- and post-intervention values.

Finally, significant correlations were observed between delta MS and PF variables (Table 4). For almost all variables considered, the results indicated that changes in MS between the pre- and post-intervention periods were significantly related to changes that occurred in the PF tests between the same period.

## 4. Discussion

This controlled clinical trial was able to demonstrate that resistance training with or without protein supplementation induced improvement in MS and PF mid-to-long-term period after RYGB. In addition, the resistance training groups showed superior results compared with the protein supplementation-only group in terms of PF.

The only similar study in the literature evaluated the effect of combined resistance training and protein supplementation interventions on MS. However, the study used a different protocol, in terms of follow-up time (18 weeks) and an earlier postoperative phase (up to 6 months). The aforementioned study found a major increase in relative lower-limb MS in the group with additional whey protein intake and supervised strength training (+0.6 [0.3 to 0.8] kg/kg body mass), measured by the one-repetition maximum protocol (1-RM), compared with the control group and additional whey protein intake-only group (+0.1 [−0.1 to 0.4] and +0.2 [0.0 to 0.4] kg/kg body mass, respectively) [11].

Several studies have reported postoperative BS improvements in terms of MS [2,11,13,26,27,28,29,30,31,32,33] or PF [2,13,26,27,28,30,32,33] after interventions involving isolated aerobic, resistance, or both types of exercise. Strength or PF enhancements have been associated with improved performance of activities of daily living in individuals with obesity [34] and reduced risk of falls, particularly related to the aging process [35]. Training benefits are important in the late postoperative period of BS, considering that weight regain typically recurs approximately 1–2 years after surgery [36] or superior period [37,38]; this result was observed in our study, where almost half of the participants regained weight. This condition has serious consequences, such as a decline in PF [39] and in the ability to perform activities of daily living, in addition to the return of obesity-related comorbidities [40].

A reliable methodology is recommended for measuring MS in clinical trials, such as isokinetic testing, which is a gold standard for evaluating maximal strength [41,42] because of its reproducibility [43], validity [44] and reliability [45]. The higher the level of MS, the lower the risk of overall mortality in the population with acute or chronic diseases [46] or even in the healthy population [47].

The 6-MWT is strongly associated with functional capacity and MS [34]. Consistent with our study, Coleman et al. also found a significant increase in the distance walked in the 6-MWT and a reduced time on the TUG test in participants with exercise intervention compared with the lack of improvement in the control group [30]. Mundbjerg et al. [26] and Stegen et al. [2] found a significant increase in the average number of repetitions on the 30-STS from baseline after supervised physical training, which is in agreement with the findings of the present study.

Some factors, such as recruitment of motor units and their activation patterns [48,49], muscle cross-sectional area, and motor unit types [49], are related to strength generation capabilities. The exercise adaptation process during resistance training involves several neuromuscular mechanisms [50]. Therefore, neuromuscular efficiency allows the generation of strength through muscle activation. Improvements in this neuromuscular mechanism occur when muscle contractions against high loads are sustained by a lower neural recruitment [51,52,53]. Muscular strength may significantly increase during the first 2–3 months in untrained individuals because of a process that is mainly associated with neural adaptations and muscle activation by neural excitation [54]. However, gains in strength in the long term are generally attributed to an increase in the cross-sectional area of the muscle fiber [55,56] and the accumulation of metabolites [57].

In our previous analysis of the same protocol, which focused on changes in body composition and resting energy expenditure, the RTP+PRO group demonstrated an increase in FFM and skeletal muscle mass compared to other groups [17]. The resistance training may have promoted muscle mass gains through the activation of the mammalian target of rapamycin (mTOR) pathway [58]. Moreover, protein supplementation could stimulate muscle hypertrophy induced by exercise through hyperaminoacidemia and high plasma bioavailability of essential amino acids [59,60,61]. Our findings showed an improvement in MS, probably attributable to nervous system adaptations in the RTP group and to muscle hypertrophy in the RTP+PRO group. It seems that the first eight weeks of resistance training performed by previously untrained individuals may not be sufficient to elicit myofiber hypertrophy, but strength gains predominantly mediated by the nervous system can be detected in the first weeks of training [51,55].

The loss of FFM can be explained in part by insufficient protein intake following BS [13]. The average amount of daily protein intake suggested by recent guidelines for patients undergoing BS is a minimum of 60 g/day [62] and up to 1.5 g/kg of ideal body weight/day to decrease FFM loss in the context of low energy intake [20]. However, these recommendations are not based on sufficient scientific evidence to support the concept of maintaining the lean body mass in the late postoperative period after BS in those engaged in a physical training program. Hence, it is important that this population, in particular, consumes high biological-value protein supplements.

Although the guidelines recommend sufficient protein intake after BS [20], patients undergoing BS may exhibit low follow-up adherence to these recommendations [63]. Possible explanations for the low adherence would be the common intolerance to protein foods [64] or protein supplementation after surgery [65], drastic reduction in the total amount of food consumed per day due to lower gastric capacity and higher hormone secretion after surgery [66], or metabolic and anatomical changes that can alter the process of digestion and absorption of macro and micronutrients, especially essential amino acids [67]. However, some factors such as lack of time, low self-discipline, lack of motivation, discomfort when exercising, and unfavorable weather are some reasons for low adherence to regular physical exercise by individuals who have undergone BS [68]. However, although the previous factors may hinder treatment after BS, participants showed satisfactory adherence to the study interventions related to physical exercise and protein supplementation (Table 2).

The resistance training protocol with adequate supervision is one of the strengths of this study, in addition to methodological rigor in all procedures and analyses involved. However, this study has some limitations, such as the lack of evaluation of baseline physical activity levels, lack of randomized allocation to the resistance training groups, and a significant loss to follow-up.

## 5. Conclusions

A supervised 12-week resistance training program improved MS and PF, regardless of protein supplementation, in individuals within 2–7 years post-RYGB, a population for whom data are rare. The results indicate that improvements in MS partially explain the noted increases in PF. Of note, improvements in functionality may indicate an enhanced ability to perform activities of daily living and, thus, an improved quality of life in this population. The results presented here support the use of resistance training as an adjunct therapy in the mid- to long term after RYGB, which is a critical stage for weight regain.

## Figures and Tables

**Figure 1 nutrients-14-00014-f001:**
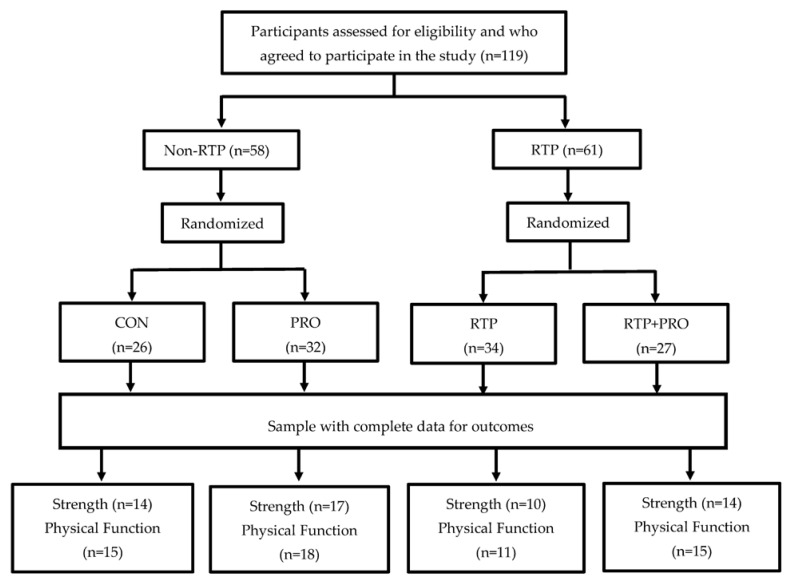
Flowchart of the participants allocation process, sample randomization, and dropout rates at each stage of the study. CON: control; PRO: whey protein supplementation; RTP: resistance training program.

**Table 1 nutrients-14-00014-t001:** Baseline sociodemographic, clinical, and anthropometric characteristics of participants in the mid-to-late postoperative period after Roux-en-Y gastric bypass, according to the intervention groups.

Variables	Completed the Study (*n* = 61)	Groups				*p* Value
		CON (*n* = 17)	PRO (*n* = 18)	RTP (*n* = 11)	RTP+PRO (*n* = 15)	
Female [*n* (%)]	54 (88.5%)	17 (100%)	16 (88.9%)	9 (81.8%)	12 (80.0%)	0.288 ^1^
Age (years)	40.2 ± 8.0	39.8 ± 7.8	40.6 ± 10.4	39.3 ± 6.9	41.0 ± 6.4	0.949 ^2^
Education level (years of study)	15.9 ± 4.0	14.3 ± 4.2	16.4 ± 4.1	16.2 ± 5.1	17.0 ± 2.6	0.256 ^2^
Years after surgery (years)	4.1 ± 1.4	3.7 ± 1.4	4.3 ± 1.4	4.7 ± 1.7	4.0 ± 1.2	0.272 ^2^
Preoperative BMI (kg/m²)	41.9 ± 5.8	41.6 ± 5.2	43.2 ± 5.8	42.5 ± 6.6	40.1 ± 5.8	0.474 ^2^
Current BMI (kg/m²)	29.8 ± 5.3	29.3 ± 4.4	30.1 ± 5.7	30.6 ± 6.2	29.4 ± 5.4	0.912 ^2^
Excess weight loss (%)	73.4 ± 21.2	75.3 ± 20.8	73.9 ± 20.9	70.7 ± 23.1	72.9 ± 22.9	0.956 ^2^
Total weight loss (%)	35.9 ± 7.1	36.5 ± 8.3	37.9 ± 6.4	34.8 ± 7.2	33.7 ± 6.3	0.342 ^2^
Weight regain ^3^ [*n* (%)]	30 (49.2%)	9 (52.9%)	8 (44.4%)	5 (45.5%)	8 (53.3%)	0.936 ^1^
Mean of weight regain (*n* = 30) (%)	17.9 ± 7.2	16.8 ± 6.6	21.3 ± 9.5	17.4 ± 7.8	15.9 ± 4.8	0.468 ^2^

^1^ Chi-square test; ^2^ One-way ANOVA; ^3^ Weight regain considered present if value > 10% of the lowest weight obtained in the postoperative period. BMI: body mass index; CON: control; PRO: whey protein supplementation; RTP: resistance training program.

**Table 2 nutrients-14-00014-t002:** Usual energy, protein intake, and study intervention adherence of individuals in the mid-to-late postoperative period after RYGB.

	Groups				*p* Value
	CON (*n* = 17)	PRO (*n* = 18)	RTP (*n* = 11)	RTP+PRO (*n* = 15)	
Food intake baseline					
kcal/day	1564 ± 296	1669 ± 306	1579 ± 241	1556 ± 393	0.707 ^1^
kcal/kg of current weight	21.0 ± 4.6	21.1 ± 4.9	19.4 ± 4.4	20.1 ± 4.3	0.746 ^1^
g protein/day	72.7 ± 4.3	72.7 ± 4.0	73.4 ± 4.0	70.1 ± 4.8	0.202 ^1^
g protein/kg of current weight	0.98 ± 0.14	0.92 ± 0.14	0.91 ± 0.22	0.93 ± 0.18	0.659 ^1^
g protein/kg of the ideal weight	1.14 ± 0.09	1.08 ± 0.09	1.09 ± 0.13	1.08 ± 0.12	0.399 ^1^
Food intake after 12-week					
kcal/day	1427 ± 408	1729 ± 428	1707 ± 521	1721 ± 517	0.187 ^1^
kcal/kg of current weight	19.5 ± 7.0	21.5 ± 6.4	20.7 ± 7.9	21.5 ± 4.9	0.776 ^1^
g protein/day	59.4 ± 14.2	81.0 ± 27.8 ^a^	67.9 ± 18.4	75.5 ± 25.9	0.041 ^1^
g protein/kg of current weight	0.80 ± 0.21	1.0 ± 0.38	0.83 ± 0.28	0.95 ± 0.32	0.203 ^1^
g protein/kg of the ideal weight	0.92 ± 0.21	1.20 ± 0.40	1.0 ± 0.26	1.16 ±0.42	0.072 ^1^
Whey protein supplementation intake					
g /day ^3^	NA	30.9 ± 2.8	NA	30.4 ± 6.0	0.740 ^2^
Whey protein adherence (%)	NA	91.9 ± 7.5	NA	92.1 ± 10.0	0.950 ^2^
Resistance Training Program					
Resistance Training Program adherence (%)	NA	NA	84.8 ± 5.2	80.0 ± 7.7	0.084 ^2^

^1^ One-way ANOVA with Tukey’s post hoc test; ^2^ Student’s *t*-test for independent samples; ^3^ Mean whey protein supplementation intake during the intervention. ^a^ *p* < 0.05 when compared with the CON group. CON: control; NA: not applied; PRO: whey protein supplementation; RTP: resistance training program.

**Table 3 nutrients-14-00014-t003:** Effect of resistance training and protein supplementation, isolated or combined, on muscle strength (MS) and physical function (PF) parameters and adherence to the resistance training program of individuals in the mid- to-late postoperative period after RYGB.

Variables	Time (Weeks)	Groups				*p* Value ^1^
		CON (*n* = 17)	PRO (*n* = 18)	RTP (*n* = 11)	RTP+PRO (*n* = 15)	
Max PT 60°/s (Nm)	0	127.91 ± 23.66	135.34 ± 40.42	149.13 ± 48.12	147.79 ± 51.39	<0.001
	12	122.28 ± 23.01	133.72 ± 42.87	162.04 ± 52.02 *^,a^	155.19 ± 50.72 *^,a^	
	Δ	−2.05 ± 4.93	−2.33 ± 5.16	11.65 ± 13.53	7.60 ± 7.69	
PT 60°/s relative to BW (Nm/kg)	0	1.70 ± 0.30	1.70 ± 0.53	1.75 ± 0.28	1.88 ± 0.33	0.002
	12	1.70 ± 0.29	1.68 ± 0.57	1.91 ± 0.30 *	1.88 ± 0.21*	
	Δ	−0.01 ± 0.05	−0.03 ± 0.07	0.12 ± 0.14	0.06 ± 0.11	
Max TW 60°/s (J)	0	439.29 ± 102.03	474.53 ± 137.69	532.27 ± 186.76	534.24 ± 195.94	<0.001
	12	415.31 ± 111.26	471.83 ± 147.26	599.04 ± 179.07 *^,a,b^	555.42 ± 188.15 *^,a^	
	Δ	−18.43 ± 48.43	−6.23 ± 16.84	75.99 ± 38.96	32.86 ± 43.44	
Max PT 180°/s (Nm)	0	85.96 ± 14.88	91.12 ± 26.64	100.00 ± 34.07	102.55 ± 38.65	<0.001
	12	83.19 ± 15.06	89.51 ± 27.57	111.53 ± 39.12 *^,a^	108.62 ± 39.20 *^,a^	
	Δ	−1.10 ± 2.59	−1.46 ± 6.74	9.28 ± 10.90	6.65 ± 4.84	
PT 180°/s relative to BW (Nm/kg)	0	1.14 ± 0.18	1.15 ± 0.40	1.16 ± 0.19	1.27 ± 0.19	0.001
	12	1.15 ± 0.16	1.13 ± 0.39	1.30 ± 0.22 *	1.31 ± 0.16 *	
	Δ	−0.01 ± 0.04	−0.03 ± 0.07	0.09 ± 0.10	0.06 ± 0.07	
Max TW 180°/s (J)	0	316.32 ± 67.02	346.70 ± 100.96	383.20 ± 120.80	385.94 ± 142.96	<0.001
	12	301.56 ± 66.91	339.40 ± 106.13	419.17 ± 115.84 *^,a,b^	417.34 ± 132.42 *^,a^	
	Δ	−7.45 ± 14.99	−10.70 ± 21.99	40.07 ± 32.57	38.31 ± 43.22	
TUG (s)	0	6.00 ± 0.75	6.23 ± 0.95	6.44 ± 0.75	5.87 ± 0.69	<0.001
	12	6.00 ± 0.69	6.06 ± 0.86	5.16 ± 0.52 *^,a,b^	5.25 ± 0.49 *^,a,b^	
	Δ	−0.02 ± 0.36	−0.17 ± 0.35	−1.20 ± 0.73	−0.62 ± 0.59	
6-MWT (m)	0	606.19 ± 63.98	577.25 ± 83.73	559.20 ± 65.12	598.49 ± 94.41	0.002
	12	579.30 ± 69.17	566.11 ± 78.01	608.70 ± 50.09 *^,b^	629.10 ± 64.39 *^,b^	
	Δ	−14.56 ± 47.05	−11.14 ± 19.71	49.81 ± 39.36	30.61 ± 75.11	
30-STS (repetitions)	0	15.29 ± 2.47	14.78 ± 3.81	14.75 ± 2.00	15.20 ± 3.19	<0.001
	12	16.00 ± 3.14	14.39 ± 3.60	17.64 ± 2.33*	17.07 ± 3.49 *^,b^	
	Δ	1.07 ± 2.43	−0.39 ± 0.98	2.70 ± 2.75	1.87± 1.40	

^1^ Two-way mixed ANOVA with repeated measures for the Time/Group interaction; ^a^ *p* < 0.05 when compared to CON; ^b^ *p* < 0.05 when compared to PRO; * significantly different from baseline in the same group based on mixed ANOVA. 30-STS: sit-to-stand test (numbers of repetitions in 30 s); 6-MWT: 6-min walk test; BW: body weight; Max: maximum; PRO: whey protein supplementation group; PT: peak torque; RTP: resistance training program group; TUG: timed-up-and-go test; TW: total work.

**Table 4 nutrients-14-00014-t004:** Correlations between delta MS and delta PF parameters of the study total sample in the mid- to late-postoperative period after RYGB.

Variables	Delta Max PT 60°/s (Nm) *n* = 53	Delta Max TW 60°/s (J) *n* = 53	Delta PT 60°/s Relative to BW (Nm/kg) *n* = 54	Delta Max PT 180°/s (Nm) *n* = 54	Delta Max TW 180°/s (J) *n* = 52	Delta PT 180°/s relative to BW (Nm/kg) *n* = 54
Delta TUG (s); *n* = 58	−0.38 (0.006)	−0.48 (<0.001)	−0.21 (0.134)	−0.48 (<0.001)	−0.44 (0.001)	−0.32 (0.019)
Delta 6-MWT (m); *n* = 57	0.44 (0.001)	0.25 (0.077)	0.39 (0.005)	0.32 (0.019)	0.28 (0.047)	0.31 (0.026)
Delta 30-STS (repetitions); *n* = 58	0.465 (0.001)	0.66 (<0.001)	0.27 (0.05)	0.45 (0.001)	0.52 (<0.001)	0.39 (0.004)

Data are presented as r or ρ, with *p*-values in parentheses for Pearson or Spearman correlations, as appropriate. 30-STS: sit-to-stand test (numbers of repetitions in 30 s); 6-MWT: 6-min walk test; BW: body weight (kg); Max: maximum; PT: peak torque; TUG: timed-up-and-go test; TW: total work.

## Data Availability

The data presented in this study are available on request from the corresponding author.

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
