# Peer review of "Resistance Training Improves Muscle Strength and Function, Regardless of Protein Supplementation, in the Mid- to Long-Term Period after Gastric Bypass"

_nutrients, 2021, doi:10.3390/nu14010014_

Round 1

Reviewer 1 Report

This introduces a risk of bias, violates the principle of randomization and does not answer the question as to what proportion of a study population would benefit from the intervention. The study would have benefited from a per protocol analysis to confirm the efficacy of the RT and protein supplementation intervention but the true question should not be whether participant complaint resistance training invokes improvements in strength and physical function, but rather, whether this population was able to comply with a program and nutritional supplementation. The study findings are not novel, the population being investigated and whether there is efficacy in supervised training and supplementation was not clearly answered. 

Reviewer 2 Report

The manuscript is well written and demonstrates this question.  It provides sufficient background and designs appropriately.  It provides sufficient background and describes methods adequately. It's not novel enough but it presents results clearly.

Round 2

Reviewer 1 Report

The authors have made a valid attempt to clarify some of the concerns but the novelty of the study is limited. The timing of the intervention post surgery is novel but the previous points regarding the challenges of implementing a nutrition and exercise intervention is the key point required to translate this to clinical practice. Thanks for clarifying your per protocol analysis but the study should have performed measures on all participants even if they declined the exercise and supplement.
